# ELAA: An Ensemble-Learning-Based Adversarial Attack Targeting Image-Classification Model

**DOI:** 10.3390/e25020215

**Published:** 2023-01-22

**Authors:** Zhongwang Fu, Xiaohui Cui

**Affiliations:** 1Key Laboratory of Aerospace Information Security and Trusted Computing, Ministry of Education, Wuhan 430001, China; 2School of Cyber Science and Engineering, Wuhan University, Wuhan 430001, China

**Keywords:** adversarial attack, black-box attack, ensemble learning, image classification, reinforcement learning, security of AI

## Abstract

The research on image-classification-adversarial attacks is crucial in the realm of artificial intelligence (AI) security. Most of the image-classification-adversarial attack methods are for white-box settings, demanding target model gradients and network architectures, which is less practical when facing real-world cases. However, black-box adversarial attacks immune to the above limitations and reinforcement learning (RL) seem to be a feasible solution to explore an optimized evasion policy. Unfortunately, existing RL-based works perform worse than expected in the attack success rate. In light of these challenges, we propose an ensemble-learning-based adversarial attack (ELAA) targeting image-classification models which aggregate and optimize multiple reinforcement learning (RL) base learners, which further reveals the vulnerabilities of learning-based image-classification models. Experimental results show that the attack success rate for the ensemble model is about 35% higher than for a single model. The attack success rate of ELAA is 15% higher than those of the baseline methods.

## 1. Introduction

With the rapid development of big data and artificial intelligence (AI) techniques, deep learning-based image-classification models are widely used in image recognition image segmentation and captioning. However, adversarial examples [1,2,3] have revealed the vulnerability of current learning based image-classification models [4]. Through adding carefully constructed perturbations indistinguishable to human vision to a raw image, an adversarial attack can be performed that forces the deep learning [5,6] classification model make wrong decisions. Such constructed images are known as adversarial samples [7]. Adversarial samples seriously affect the usability of existing AI systems. For example, in the field of self-driving cars, if the traffic sign [8,9] is perturbed and misclassified by the AI system, it may cause serious consequences, such as traffic accidents [7]. By using the adversarial patches in the physical world [10,11,12,13], the attacker can fool a recognition model without accessing the digital input to it, making them an emerging threat to deep learning applications, especially to face recognition [14,15] systems in security-sensitive scenarios.

Current research on adversarial attacks is mainly focused on evasion attacks. Evasion attacks are to escape the prediction results of the classification model by constructing adversarial examples. We focus on evasion attacks, since the input images are easy to obtain in most real world cases. Evasion attacks can be divided into white-box attacks and black-box attacks [16,17,18,19] according to the different access of the attacker to the target model [4]. White-box attacks require the attackers to have full access to the target model. They need to know the model’s architecture, parameters, and gradients, which is demanding in many cases. Black-box attacks rarely have or do not require to the inside information of the target model, which is more practical. Thus, we focus on black-box attacks. According to the purpose of the attack, evasion attacks can be divided into targeted attacks and untargeted attacks. A targeted attack refers to model misclassifying the constructed adversarial sample to a specific targeted category. If the target model misclassifies the adversarial sample into any category other than the original category, it is known as the untargeted attack [7]. The research in this article belongs to the category of untargeted attacks to maximize perturbed consequences.

The main contributions of this paper are as follows:(1)Based on AutoAttacker, a black-box adversarial-attack framework based on reinforcement learning, a new black-box adversarial sample attack model is proposed ELAA—an adversarial sample attack model targetting image classification based on ensemble learning. Adversarial samples can be generated without knowing the internal information of the attacked network, such as structure and weight.(2)BAGGING method is adopted for integrated learning of reinforcement-learning-based base learning, and voting combinations effectively strengthen the advantages of each base learning.(3)Taking the attack on the classical image-classification model ResNet as an example, the experiment results show a significant attack effect. The attack success rate with ensemble learning is about 35% higher than that with a single learning model. The attack success rate of ELAA is 15% higher than any of the baseline methods.

The remainder of this paper is structured as follows: In Section 2, we introduce related works on evasion attacks. In Section 3, we propose a novel black-box adversarial attack model named ELAA. Experimental results are discussed in Section 4. Finally, conclusions and future works are given in Section 5.

## 2. Related Works

In this section, we introduce some adversarial attack methods [20,21,22,23,24,25,26] that are closely related to this paper. The discovery of adversarial samples began with the exploratory research on the interpretation of deep learning models in image classification by researchers such as Szegedy in 2013 [20]. They constructed adversarial samples through L-BFGS attacks under white-box conditions and successfully made the deep learning model identify a panda as a gibbon incorrectly. In recent years, research on adversarial attacks for deep learning models in image classification has attracted more and more attention worldwide.

In reference [26], the authors use the idea of zero-order optimization and propose the adversarial attack ZOO, which estimates the zero-order target gradient by querying the target neural network, and then uses the estimated rise to optimize the loss function of C&W [27] to generate adversarial samples. Subsequently, inspired by references [28,29], the authors of [30] estimated the gradient of the target model by searching the density of the normal distribution and then used projected gradient descent (PGD) to minimize the objective function to generate adversarial samples. The idea of PGD was further studied and improved in reference [31] specifically. Compared to directly minimizing the objective function in PGD, the goal in reference [31] is to find a distribution that fits the vicinity of the original data, so its realization may be adversarial. In another study, the authors of reference [32] observed that the gradient used in adversarial samples is highly correlated with time and cross-data. Therefore, if there is a priori knowledge about the gradient, the number of queries to attack the black-box model can be reduced. The method in [33] is an intuitive black-box adversarial attack construction method. It is said that this combination can reduce the probability of correct identification of the label for a given gradient direction and step length. In reference [34], the authors propose two types of ensemble-based black-box attack strategies called SCES and SPES to establish a substitute model and generate the adversarial examples for the substitute model approximating the target system.

In addition, there are also black-box attack methods based on generative adversarial networks (GAN) [35,36]. Tsingenopoulos et al. proposed a reinforcement learning-based black-box adversarial attack framework called AutoAttacker [35]. The agent can learn to operate by querying the black-box model to extract the underlying decision behavior and effectively break it. This attack method is the first framework to use reinforcement learning. It is robust to common defenses such as adversarial training [37,38] because it assumes nothing about the structure of the underlying function. However, the attack success rate of the AutoAttacker on the MNIST dataset is only 73.4%. Reference [39] proposes a novel modelization of the process of learning an attack policy as a multi-objective Markov decision process with two objectives. Sun Yiwei et al. proposed a novel reinforcement learning method for node injection poisoning attacks (NIPA) to sequentially modify the labels and links of the injected nodes without changing the connectivity between existing nodes [40]. There are patch-based methods of black-box attack methods. PatchAttack [41] induces misclassifications by superimposing small textured patches on the input image. RLAB, reinforcement learning for adversarial black-box attack [42], is based on selective Gaussian noise distortion of specific fixed-size square patches in the image with the images split into multiple patches of size n × n. Motivated by AutoAttacker, we propose a novel black-box attack model utilizing ensemble learning which combines models under reinforcement learning to further improve the attack success rate.

## 3. Proposed Method

### 3.1. Basic Idea

At present, white-box image classification attack methods are not suitable for the actual attack scene because they need to know the attack model’s structure, weight, and other internal information. Meanwhile, in some machine-learning-based black-box attack methods, it is not easy to train a stable black-box attack model with sound effects for all types of labels, and sometimes, tendentious models are obtained. Can we bring together the advantages of these diversity models and circumvent the disadvantages under a unified framework? Ensemble learning is a learning method of optimization that works by training several base learners (models) through specific rules and then taking a combination strategy to form a stronger learner with better performance to improve the attack effect. Thus, we propose a new black-box attack model named ELAA which is an ensemble-learning-based adversarial attack targeting image-classification models.

### 3.2. Assumptions and Definitions

This paper follows the same black-box case in [35]: we have no knowledge of the attacked model, and the remaining job for an attacker is to submit queries to the attacked model and record the outputs.

Given an image dataset *X* and an image classifier *R*: $x\to \left\{1,2,\cdots ,n\right\}$, where x∈X, an untargeted attack aims to add a perturbation δ to *x* to compute $x^{{}^{\prime }}$, such that $R\left(x\right)\neq R\left(x^{{}^{\prime }}\right)$. Then, *x* is called an original image, and $x^{{}^{\prime }}$ is called an adversarial sample. The perturbation δ is chosen to be sufficiently small to be invisible to human eyes. In the context of adversarial attacks, the distance metric is the $L_{p}$ norm of the difference between the original image *x* and the adversarial image $x^{{}^{\prime }}$. In this paper, $L_{2}$ norms are used to quantify the amount of perturbation added to create an adversarial sample as in formula (1).
(1)$${\|x,x^{{}^{\prime }}\;\|}_{2}=\sqrt{\sum_{i=1}^{n}{\left(x_{i}-x_{i}^{{}^{\prime }}\right)}^{2}}$$

### 3.3. Overview of Proposed Model

In general, the proposed model ELAA is an adversarial attack model based on ensemble learning and reinforcement learning, and it can be divided into two parts. The first part includes several base learners based on reinforcement learning, used to generate image-classification-adversarial samples. The second part is a bagging-based ensemble learning framework used to combine the base learners. The architecture of ELAA is illustrated in Figure 1.

The basic process of ELAA is:

Step 1: Sampling for n rounds. Based on the bagging algorithm, the original training set *X* is randomly sampled for *n* rounds to form *n* training subsets $X_{1},X_{2\cdots },X_{n}$.

Step 2: Training in parallel. For each training subset $X_{i}$, a base learner $B_{i}$ is trained with reinforcement learning. Each base learner takes the following action to generate a new perturbation and an in-process sample through the return of the deep learning classifier to the agent. Then, base learner *n* can be obtained.

Step 3: Voting. Based on the voting strategy, *n* base learners are combined to produce the final adversarial sample set.

The two parts of ELAA are introduced in detail.

### 3.4. Base Learner with Reinforcement Learning Agent

In this paper, the reinforcement learning agent is taken as the base learner of ensemble learning. Reinforcement learning (RL) consists of environment, action, and state [35]. Agent draws rewards by taking action in a given environment, and then rewards are maximized through the learning process. The core of RL is to obtain an optimized policy π, which maximizes expectations *E* of rewards. Denote step as *t* and discount factor as γ. An optimized policy can be formed through formula (2):(2)$$\pi^{*}=\underset{\pi }{\mathrm{argmax}}E\left[\sum_{t\geq 0}\gamma^{t}r_{t}|\pi \right]$$

Prevalent RL algorithms are provided in Table 1, and the one utilized in this paper is the actor–critic (AC) algorithm.

AC algorithm takes advantage of policy-based RL and value-based RL, which can be used in continuous action space and updated within a single step. AC is composed by the actor network and the critic network. The actor network produces actions tokens in the given environment. At the same time, the critic network decides on the quality of actions, and then the actor network optimizes the next action guided by the critic network [35]. The actor–critic algorithm in the proposed model is described in Algorithm 1.
**Algorithm 1:** Actor-Critic in the proposed modelInput: Iteration *T*, time step α, discount factor γ, hypermeter for policy network θProcess:Initialize observations of states $\varphi \left(s_{1}\right),\varphi \left(s_{2}\right),...,\varphi \left(s_{n}\right)$for i=1,2,...,T:$R_{i}$,$\varphi^{\prime }\left(s_{i+1}\right)$,$A_{i+1}$ = Actor($\varphi (s_{i})$,$A_{i}$)$V(s_{i})$,$V^{\prime }\left(s_{i}\right)$ = Critic($\varphi (s_{i})$,$\varphi^{\prime }\left(s_{i}\right)$)Update TD Error by$\delta \leftarrow R_{i}+V^{\prime }\left(s_{i}\right)-V\left(s_{i}\right)$Update Critic by$\omega \leftarrow \omega +\beta \delta V(s_{i})$Update Actor by$\theta \leftarrow \theta +\alpha {▿}_{\theta }\log\pi_{\theta }\left(S_{t},A_{i}\right)\delta$Update State by$\varphi \left(s_{i}\right)=\varphi^{\prime }\left(s_{i+1}\right)$end forOutput: ω,θ

The basic structure of RL-based base learner $B_{i}$ used in the proposed model is shown in Figure 2. For each base learner $B_{i}$, the training process based on reinforcement learning is as follows:

Step1: Input the training subset $X_{i}$ and initialize states $s_{1},s_{2},\cdots ,s_{n}$.

Step 2: Save all Q-value into DNN, and fit the Q-value with a neural network.

Step 3: Each agent saves the buffer, including a tuple of $env(s_{1},A_{i},reward,s_{i+1})$.

Step 4: With sampling tuples from saved buffers, actions of the largest Q-value are returned by the Q-network.

Step 5: The state is updated by the action, and a DNN calculates the Q-values of all actions in the current state.

**Figure 2 entropy-25-00215-f002:**
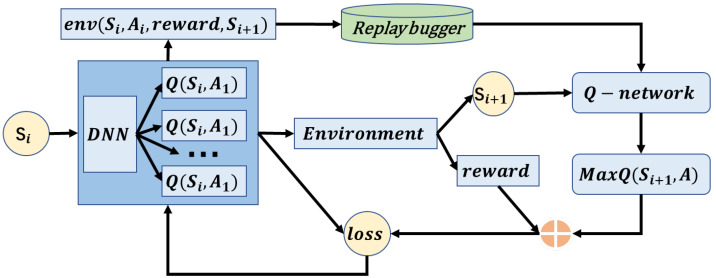
The basic structure of a single base learner.

In detail, we take a 300-dimensional continuous state space consisting of a 2-dimensional convolution feature map of the input image and the last layer of the target network. Action space is defined as the direct perturbations [43] of the input image. Taking the MNIST dataset as an example, the input dimensions are 28 × 28, and the corresponding action space is a 784-dimensional vector, where the actor network is defined as follows: input of a 300-dimensional state-space vector and two hidden layers whose size is 512. The input of the critic network consists of 384-dimensional state space and a 784-dimensional action vector. The hidden layers are the same as in the actor network. The output is a single neuron containing the Q-values. Since RL agents cannot learn from human experience, the design of the reward function is critical to the performance of RL agents. We refine the reward function of the adversarial model: Aside from the maximal reward for evasion, we consider top-1 to top-5 classes and confidences’ transformation. When top-1 confidence decreases or top 2–5 confidences increase, we give rewards accordingly to their specific transformation, avoiding the RL agent’s inefficacy caused by a sparse reward function. Our experimental results in attacking the MNIST dataset also demonstrate the improvement of our innovation.

### 3.5. Ensemble Model Based on Bagging

Ensemble learning is a machine learning paradigm, and it combines some base learners trained to solve the same problem to obtain an ensemble model with better results [44]. There are three commonly used structures of ensemble learning: bagging, boosting, and stacking. A bagging structure is used here. Under the framework of bagging, base learners are homogeneous and can be trained in parallel. Bagging structure is a relatively easy way to improve the effectiveness of an existing method, and it also has high learning efficiency.

Figure 3 illustrates the basic structure of the ensemble model based on bagging in ELAA.

The ensemble learning model based on bagging uses a statistical technique named bootstrapping to generate samples of size *K* from an initial dataset of size *M* by randomly drawing with replacement *K* observations.

In brief, the process of the ensemble with bagging in ELAA is given as follows:

Firstly, get n training subsets from the training dataset, X, through bootstrapping sampling with the replacement for *n* rounds. Secondly, train an attack model $B_{i}$ based on subset $X_{i}$ in parallel, and then get *n* base learners. Thirdly, combinatorial optimization of these *n* base learners is conducted by a combining strategy, and the plurality voting method is used here. Finally, the optimized result is taken as the final result.

The algorithm of bagging in the proposed model is described in Algorithm 2.
**Algorithm 2:** Bagging algorithm in the proposed modelInput: Datasets$\left(X,Y\right)=\{\left(x_{1},y_{1}\right),\left(x_{2},y_{2}\right),\left(x_{3},y_{3}\right),...,\left(x_{m},y_{m}\right)\}$,$x_{j}\in X,x_{j}$ is an image, $y_{j}\in Y,y_{j}$ is the label of $x_{j}$:Base learner algorithm:Iterations *n*.Process:for i=1,2,3,⋯,nSampling randomly from training set *X* using bootstrapping and $X_{i}$ can beobtained, $X_{i}$ is a subset of the training set *X*:Training base learner $B_{i}$ with RL ondataset $X_{i}$;end forOutput: The results of each base learner $B_{i}$ are combined into a final result by aplurality voting strategy,$F\left(x\right)=argmax_{y\in Y}\sum_{i=1}^{n}\prod \left(f_{i}\left(x\right)\neg y\right)$

## 4. Experiments and Analysis

### 4.1. Target Model and Datasets

The attacked model used in the experiments is ResNet-152, one of the best deep neural networks in the image classification tasks. The classification accuracy rate of the attacked model is over 98% in non-attacking cases in the experiments. All the experiments are based on black-box attacks, and only black-box access is allowed for the ResNet-152 model. The internal structure and other information of ResNet-152 would not be used in attacks.

The MNIST dataset, an image dataset that is often used to evaluate adversarial attack performance, was selected for the experiments. The MNIST dataset is from the National Institute of Standards and Technology of the United States. It is a handwritten digital dataset, including 60,000 pieces in the training set and 10,000 pieces in the testing set, and each image is 28 × 28 and grayscale [35]. Bootstrapping sampling was used during training to sample the training set for n rounds, and n training subsets were obtained, each of which contained 50,000 images. One thousand pieces were selected from the testing set to evaluate the effectiveness of attacks.

### 4.2. Experimental Results and Analysis

#### 4.2.1. Performance of Ensemble Learning of ELAA on MNIST

To evaluate the attack performance of the ensemble method based on bagging in ELAA, we used different numbers of base learners. Setting n=10, for each base learner trained on n different subsets by RL, the attack performance of individual base learner $B_{i}$ was tested against the ResNet-152 model on the MNIST test dataset. The attack success rates (SR) of individual base learners (BL) are shown in Figure 4 below.

Figure 4 shows that the attack success rates of individual base learners can be more than 50% but cannot reach 60%. This indicates that the attack success rates of the n individual base learners trained by RL on different subsets are low, and the attack performance is not good enough.

Figure 5 shows the attack success rates of ELAA after integrating different numbers of base learners, where the integration of two base learners is designated as E2, the integration of three base learners is designated as E3, and so on.

As shown in Figure 5, the integration of two base learners can increase the attack success rate, and the attack success rate approached 70%. With the increase in base learners, the attack success rate gradually increased. When the number of base learners was 10, the attack’s success rate became close to 90%—nearly 35% higher than that of a single one.

#### 4.2.2. Performance of Ensemble Learning of ELAA on CIFAR-10

For CIFAR-10, we used a ResNet-50 target model, and we compared it to SimBA, AutoZOOM, and CompleteRandom. We use the following shorthand in the results below: SR (success rate), QA (queries’ average), and L2 (average L2 norm of successful perturbations).

We use SimBA-CB to refer to the results of the Cartesian basis paper’s results [45] and SimBA-DCT for the DCT paper’s results [45]. In the cases, for which the AutoZOOM paper [46] provided data, we only compare our method to AutoZOOM-BiLIN, the version of the attack which requires no additional training and data.

In Table 2, all the attacks are shown to have achieved a 100% success rate in the CIFAR-10 experiments, with the sole exception of CompleteRandom, which only got 69.5%. The success rate of CompleteRandom (69.5%) and its low average query count (161.2) are surprisingly good results for the trivial nature of the method, outlining once again the lack of robustness in complex image classifiers. However, this come at the cost of an average QA that is roughly two times higher and of an average L2 that is roughly 2.5 times higher in comparison with the ELAA results. All the comparative indicators show that ELAA has better performance.

#### 4.2.3. Comparison of Attack Performance between ELAA and AutoAttacker

AutoAttacker is a reinforcement-learning-based attack method presented in reference [35], for which the attack success rate was 73.4% on the MNIST dataset. According to the comparison results in Figure 6, the attack success rate of a single base learner in the ELAA model is less than that of AutoAttacker. Still, it would exceed that of AutoAttacker when ensembling three base learners. After ensembling more base learners, the attack success rates increase steadily. The attack success rate approached 90% when ensembling 10 base learners, which is nearly 15% higher than the attack success rate of AutoAttacker. This verifies the effectiveness of the attack method presented in this paper.

#### 4.2.4. Adversarial Examples

Figure 7 shows some original images and corresponding adversarial examples generated on the MNIST dataset by the ELAA model. The first and the third rows are the adversarial images, and the second and the fourth rows are the original images.

Furthermore, the ELAA model was not only effective on the MNIST dataset but also feasible for the CIFAR10 dataset (also often used to evaluate adversarial attack performance) in further experiments, which reflects the generality of this model. Figure 8 shows some original images and corresponding adversarial examples generated on the CIFAR10 dataset by the ELAA model. The first and the third rows are the adversarial images, and the second and the fourth rows are the original images.

## 5. Conclusions

The security of deep neural network models is an urgent problem in current AI systems. Adversarial attacks seriously affect the security of the current deep neural network models for image classification. This paper proposed a novel black-box adversarial attack model named ELLA, an ensemble-learning-based adversarial attack model targeting image-classification models. With the bagging method, the combined optimization of multiple base learners is performed with reinforcement learning in the ELAA model. Experimental results on the ResNet model and MNIST dataset showed that the ELLA model is effective and outperforms the baseline method. Since it is a black-box model and does not rely on any specific model architecture, the generative method of adversarial samples proposed in this paper is also applicable to other neural networks, such as DNNs and CNNs. Meanwhile, the dataset testing could also be extended to the CIFAR10 or other datasets. A future step will be to explore a defense method against the proposed attack model.

## Figures and Tables

**Figure 1 entropy-25-00215-f001:**
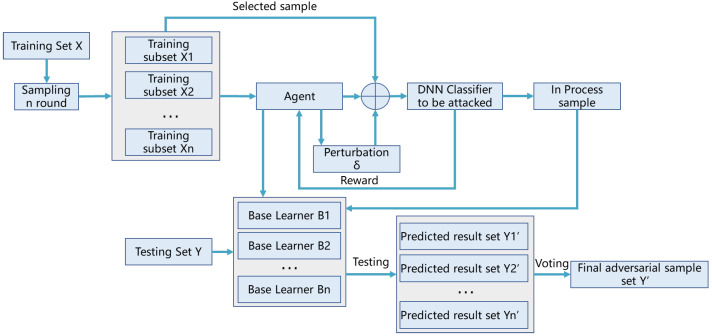
The architecture of the proposed model, ELAA.

**Figure 3 entropy-25-00215-f003:**
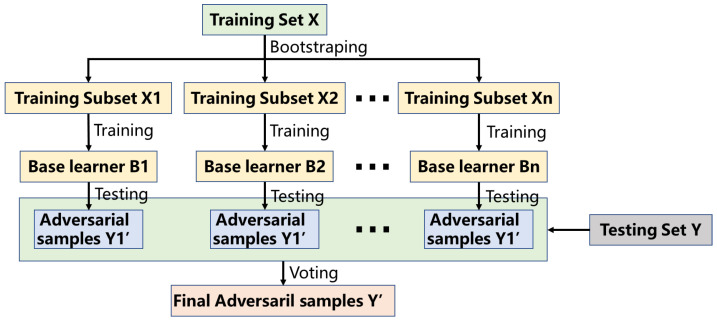
The ensemble model based on bagging in ELAA.

**Figure 4 entropy-25-00215-f004:**
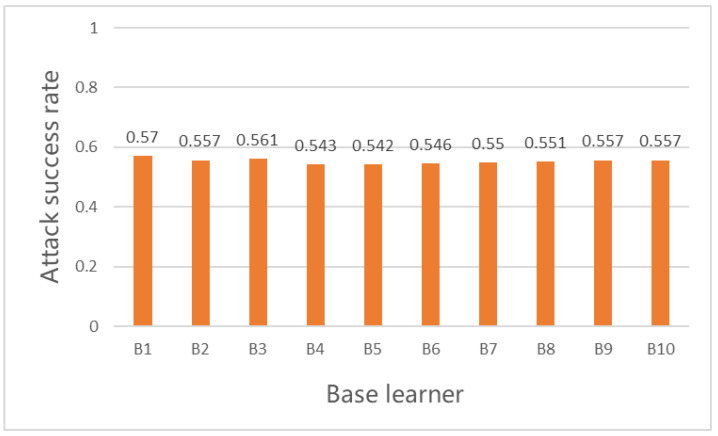
The attack success rate of each base learner.

**Figure 5 entropy-25-00215-f005:**
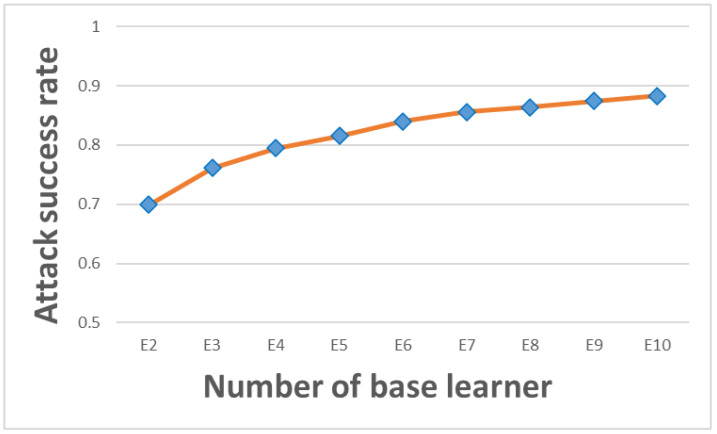
The attack success rate after integration of different numbers of base learners. The orange line represents the change trend of attack success rate of ELAA after integrating different number of base learners. The blue diamond represents attack success rate.

**Figure 6 entropy-25-00215-f006:**
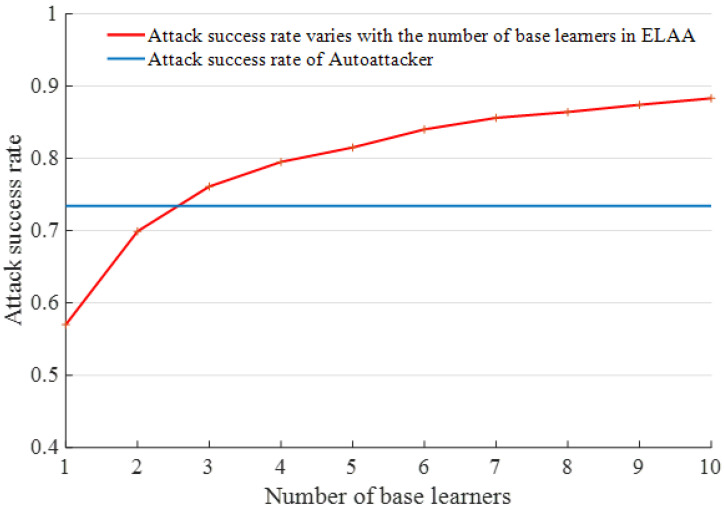
Comparison of attack success rate between ELAA and AutoAttacker.

**Figure 7 entropy-25-00215-f007:**
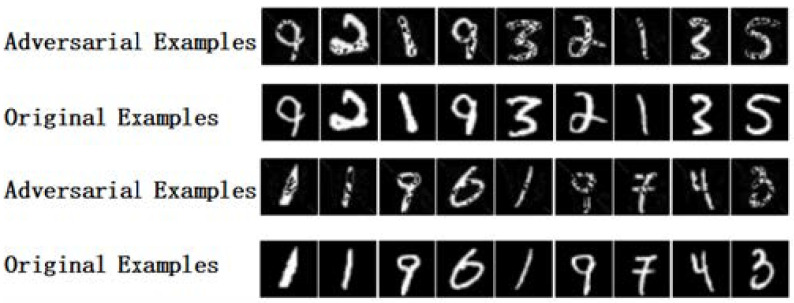
Comparison between the counter sample image and the original sample image from the MNIST dataset.

**Figure 8 entropy-25-00215-f008:**
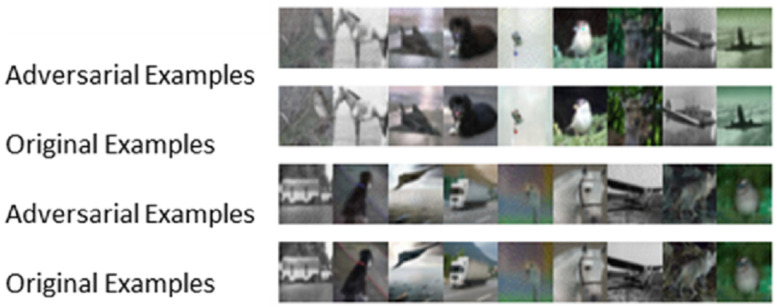
Comparison between the counter sample image and the original sample image on the CIFAR10 dataset.

**Table 1 entropy-25-00215-t001:** Prevalent RL algorithms. It mainly includes Q-learning, Sarsa, Policy-Gradient, Deep Q Network, and Actor-Critic. The table shows their characteristics. It mainly includes model-free (not based on environment), policy-based (policy-based reinforcement method), Value-Based(value-based reinforcement method), On-Policy, and Off-Policy(whether to use the samples generated by the current Policy).

Algorithms	Model-Free	Policy-Based	Value-Based	On-Policy	Off-Policy
Q-Learning	✓		✓		✓
Sarsa	✓		✓	✓	
Policy-Gradient	✓	✓			
Deep Q Network	✓		✓		✓
Actor-Critic	✓	✓	✓		

**Table 2 entropy-25-00215-t002:** CIFAR-10 results.

	SR	QA	L2
SimBA-CB	100%	322	2.04
SimBA-DCT	100%	353	2.21
AutoZOOM-BiLIN	100%	85.6	1.99
CompleteRandom	69.6%	161.2	3.89
ELAA	100%	83.91	1.53

## Data Availability

The MNIST dataset, an image dataset that is often used to evaluate adversarial attack performance, was selected for the experiments. The MNIST dataset is from the National Institute of Standards and Technology of the United States. It is a handwritten digital dataset, including 60,000 pieces in the training set and 10,000 pieces in the testing set, and each image is 28 × 28 and grayscale. The CIFAR-10 or other datasets could also be used. CIFAR-10 is a small dataset collated by Hinton students Alex Krizhevsky and Ilya Sutskever for the identification of pervasive objects. It has a total of 10 categories of RGB color images, airplane, automobile, bird, cat, deer, dog, frog, horse, ship, and truck. The size of the images is 32 × 32, and the dataset consists of 50,000 training images and 10,000 test images.

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
