# Peer review of "ELAA: An Ensemble-Learning-Based Adversarial Attack Targeting Image-Classification Model"

_entropy, 2023, doi:10.3390/e25020215_

Round 1

Reviewer 1 Report

The paper uses an ensemble of RL based approaches to generate adversarial samples for image classification models
The authors have used AC based RL learning algorithm to generate adversarial samples.

Related work:

The related work is weak.  Although they have addressed adversarial attacks in general and some of GAN based approaches.
There are numerous reinforcement learning based adversarial attacks in the literature:

1. Measuring Robustness with Black-Box Adversarial Attack using Reinforcement Learning

@inproceedings{
sarkar2022measuring,
title={Measuring Robustness with Black-Box Adversarial Attack using Reinforcement Learning},
author={Soumyendu Sarkar and Sajad Mousavi and Ashwin Ramesh Babu and Vineet Gundecha and Sahand Ghorbanpour and Alexander K Shmakov},
booktitle={NeurIPS ML Safety Workshop},
year={2022},
url={https://openreview.net/forum?id=Lj8fj0ECPv}
}

2. Patch Attack

inproceedings{yang2020patchattack,
  title={Patchattack: A black-box texture-based attack with reinforcement learning},
  author={Yang, Chenglin and Kortylewski, Adam and Xie, Cihang and Cao, Yinzhi and Yuille, Alan},
  booktitle={European Conference on Computer Vision},
  pages={681--698},
  year={2020},
  organization={Springer}
}

3. Adversarial attacks on graph neural networks via node injections: A hirerchical reinforcement learning approach

@inproceedings{sun2020adversarial,
  title={Adversarial attacks on graph neural networks via node injections: A hierarchical reinforcement learning approach},
  author={Sun, Yiwei and Wang, Suhang and Tang, Xianfeng and Hsieh, Tsung-Yu and Honavar, Vasant},
  booktitle={Proceedings of the Web Conference 2020},
  pages={673--683},
  year={2020}
}

4. Learning adversarial attack policies through multi-objective reinforcement learning

@article{garcia2020learning,
  title={Learning adversarial attack policies through multi-objective reinforcement learning},
  author={Garc{\'\i}a, Javier and Majadas, Rub{\'e}n and Fern{\'a}ndez, Fernando},
  journal={Engineering Applications of Artificial Intelligence},
  volume={96},
  pages={104021},
  year={2020},
  publisher={Elsevier}
}

Concerns:

Approach:

While this approach seems very effective for a MNIST dataset with the input dimension of 28x28, given the nature of
RL techniques, the action space will increase with increase in the dimension of the data, eg. Imagenet

Furthermore, the state space is too large for the given policy network and the learning algorithm to converge.

There is no details about the kind of distortions that is being added to the input image.  

Evaluations:

There are evaluations provided only for the MNIST dataset for the ensemble based approach.  But some of the important factors for an adversarial attacks are following,

Average success rates, Linf, L1, L2 from the perturbations, average number of queries.  More information on these are required to show the effectiveness.  

Reviewer 2 Report

Summary

This paper proposed a new untargeted black-box attack based on ensembling with reinforcement learning based learners. The paper provides detailed algorithms on how to design the base learner and how to ensemble with bagging. Experimental comparisons against Autoattacker is given on MNIST and ResNet-152.

Strength

  • The proposed method is interesting. RL-based and ensemble-style blackbox attacks are not well explored in the literature.

  • Results against Autoattacker presented in figure 6 shows quite promising results.

Weakness

  • I am mostly concerned about the evaluation. The current evaluation is not sufficient. The paper did not compare with any of existing SOTA blackbox attacks like square attacks. I understand it might be hard to outperform them but giving a clear comparison is very helpful to understand how the proposed attack performs.

  • MNIST is for now far too simple for robustness evaluation. At least CIFAR10 should be considered as the main benchmark. Comparisons over standard robustness benchmark like RobustBench should make results much more convincing.

  • The presentation should be improved. The insights of the proposed methods and how they addressed the previous limitations should be emphasized in the introduction. Also it is meaningless to mention poisoning attack at all.

Round 2

Reviewer 1 Report

Thanks for addressing my major concerns, although,

just for your introduction, your major contributions, point 1 and 2 are not understandable.  Please refine the language to convey what you wanted to in a much readable way.

Rest of the concerns are addressable.

Reviewer 2 Report

Thank the authors for the detailed response and revision. Most of my concerns have been addressed. Large scale evaluation on RobustBench as well as comparisons with more SOTA adversarial attacks would be a plus but could be extended as future work.